# Exploring the Role of PI3P in Platelets: Insights from a Novel External PI3P Pool

**DOI:** 10.3390/biom13040583

**Published:** 2023-03-24

**Authors:** Abdulrahman Mujalli, Julien Viaud, Sonia Severin, Marie-Pierre Gratacap, Gaëtan Chicanne, Karim Hnia, Bernard Payrastre, Anne-Dominique Terrisse

**Affiliations:** 1Institut des Maladies Métaboliques et Cardiovasculaires (I2MC), INSERM UMR-1297, Université Paul Sabatier, F-31432 Toulouse Cedex, France; 2Laboratoire d’Hématologie, Centre de Référence des Pathologies Plaquettaires, Centre Hospitalier Universitaire de Toulouse Rangueil, F-31432 Toulouse Cedex, France

**Keywords:** phosphoinositides, PI3P, outer leaflet of the plasma membrane, platelets, endocytosis, α-granules

## Abstract

Phosphoinositides (PIs) play a crucial role in regulating intracellular signaling, actin cytoskeleton rearrangements, and membrane trafficking by binding to specific domains of effector proteins. They are primarily found in the membrane leaflets facing the cytosol. Our study demonstrates the presence of a pool of phosphatidylinositol 3-monophosphate (PI3P) in the outer leaflet of the plasma membrane of resting human and mouse platelets. This pool of PI3P is accessible to exogenous recombinant myotubularin 3-phosphatase and ABH phospholipase. Mouse platelets with loss of function of class III PI 3-kinase and class II PI 3-kinase α have a decreased level of external PI3P, suggesting a contribution of these kinases to this pool of PI3P. After injection in mouse, or incubation ex vivo in human blood, PI3P-binding proteins decorated the platelet surface as well as α-granules. Upon activation, these platelets were able to secrete the PI3P-binding proteins. These data sheds light on a previously unknown external pool of PI3P in the platelet plasma membrane that recognizes PI3P-binding proteins, leading to their uptake towards α-granules. This study raises questions about the potential function of this external PI3P in the communication of platelets with the extracellular environment, and its possible role in eliminating proteins from the plasma.

## 1. Introduction

Phosphoinositides (PIs) are glycerophospholipids containing a hydrophilic inositol head group that can be reversibly phosphorylated at the positions 3, 4, and 5 by specific lipid kinases and phosphatases, resulting in 7 distinct PIs [1,2]. These lipids play important roles in signal transduction, cytoskeleton remodeling, and vesicular trafficking [2,3,4,5]. They act as precursor of second messengers or as direct lipid mediators in membranes facing the cytosol to regulate the spatiotemporal organization of protein complexes through binding to specific domains of effector proteins [2,5,6,7,8,9,10].

Imaging studies using fluorescently-tagged PIs probes have shown that these lipids are not uniformly distributed throughout cellular membranes [1,2,3,9,10]. For example, phosphatidylinositol 4,5-bisphosphate (PI(4,5)P_2_) is mainly found in the plasma membrane while PI4P is enriched in the Golgi apparatus and is also present at the plasma membrane [9,10]. PI3P has been primarily described in early endosomes, where it controls several steps of the endocytosis process via the recruitment of proteins containing a FYVE (Fab1, YotB, Vac1p, and EEA1) domain [11]. Additionally, PI3P is important in autophagy through the binding and recruitment of proteins such as WIPI2 to the autophagosomal membrane [12]. PI3P is also involved in the activation of the NADPH oxidase by recruiting PX (Phox homology) domain-containing proteins such as p40Phox [13]. In addition to this, other locations of PI3P have been identified, including the inner leaflet of the plasma membrane in adipocytes stimulated by insulin [14]. This highlights the fact that the intracellular synthesis of PI3P is tightly regulated in terms of both space and time, and that maintaining a steady pool of PI3P is essential for preserving cell homeostasis [15].

Historically, PIs have been primarily studied on the membrane leaflets facing the cytosol, where they can interact with effector proteins or serve as a substrate for enzymes such as kinases, phosphatases, and phospholipases. However, a few reports suggest that some PIs can be present in the outer leaflet of the plasma membrane, with the potential to integrate signals from and to the extracellular microenvironment. Interestingly, a pool of PI3P has been found in the outer leaflet of plant cell plasma membrane, where it facilitates the entry of pathogen effector proteins with PI3P-interacting domains such as the RxLR motif [16,17]. The presence of a PI3P pool in the outer leaflet of animal host cells has also been suggested [16]. Recently, PI(4,5)P_2_ has been identified in the outer leaflet of mammalian cell plasma membrane, where it may regulate adhesion and motility [18]. Similarly, a pool of PI(3,4,5)P_3_ has been observed flipping in the outer leaflet of apoptotic cell plasma membrane as a “eat-me” signal recognized by CD14+ phagocytes [19]. These findings suggest that, in addition to inward-facing PIs, which are well-known signaling molecules, PIs in the outer leaflet of plasma membranes may have the potential to integrate signals from and to the extracellular microenvironment. The PIs metabolism is particularly active and important in blood platelets [20,21,22], providing the opportunity to investigate these lipids in an easily obtainable primary cell.

In this study, we used various PI-binding domains and recombinant PIs metabolizing enzymes to investigate the presence of these lipids in the outer leaflet of the plasma membrane of human and mouse platelets. We uncovered a new pool of PI3P in the outer leaflet of the platelet plasma membrane and found that specific binding of proteins to this pool induced their uptake towards α-granules ex vivo and in vivo.

## 2. Materials and Methods

Reagents and antibodies: Sepharose 4B beads were purchased from GE Healthcare, reduced gluthathione from Sigma–Aldrich (St. Louis, MO, USA), and Di-C16-^bodipy^PI3P from Echelon Biosciences (Salt Lake City, UT, USA). Fluorescently-tagged Fibrinogen/Oregon Green was purchased from ThermoFisher Scientific (Waltham, MA, USA). Anti-GST antibody (B-14, sc-138) was purchased from Santa Cruz Biotechnology (Dallas, TX, USA), HRP secondary antibody from Promega (Madison, WI, USA), and anti-PI3P (Z-P003) from Echelon Biosciences (Salt Lake City, UT, USA).

Plasmids and recombinant proteins: pGEX vectors encoding GST-FYVE (Hrs), GST-PH (GRP1), GST-PH (FAPP1), and GST-PH (PLC-δ1) were gifts from M. Lemmon (University of Pennsylvania, Philadelphia, PA, USA). PHD2x (ING2) was from Or Gozani (Stanford, CA, USA). PI-binding domains were first subcloned in a pmCherry-C1 (Clonetech, Mountain View, CA, USA) and mCherry PI-binding domains were then subcloned in a pGEX-4T-1 (Cytiva, Marlborough, MA, USA) vector using the restriction enzymes and the primers listed in Appendix A. pGEX-p40(Paa1-148) (RRID:Addgene_19008) and pGEX-p40R57Q PX (RRID:Addgene_19009) were gifts from Michael Yaffe. The plasmids expressing AVRL567 were kindly provided by Dr. B.M. Tyler (Oregon state university, Corvallis, OR, USA). The plasmids expressing ABH and its catalytically inactive mutant ABH_SA_ were kindly provided by Professor Karla Satchell (Chicago, CA, USA). Mutated probes were prepared by PCR mutagenesis. Recombinant proteins were expressed overnight at 18 °C using 1 mM IPTG in *Escherichia coli* strain BL21(DE3) and purified by affinity chromatography using glutathione Sepharose 4B beads according to the manufacturer’s instructions (GE Healthcare, Chicago, IL, USA) in 50 mM Tris at pH 8.0 and 100 mM NaCl, as previously described [23]. The GST-tagged PX domain of p40 and its inactive mutant were labelled by conjugation of amine-reactive Alexa Fluor 594 and 488 carboxylic acid succinimidyl ester (ThermoFisher Scientific (Waltham, MA, USA)) according to the manufacturers protocol. The recombinant ^GFP^AVR L567 protein was produced in *Escherichia coli* strain BL21 (DE3) as GST-tagged fusion as described [16]. The specificity of the probes was tested using Fat blot or PLIF assay [23] and confirmed a selective binding of FYVE (Hrs) to PI3P, PH (FAPP1) to PI4P, PHD2x-(ING_2_) to PI5P, PH (PLCδ1) to PI(4,5)P_2_ and PH-(GRP1) to PI(3,4,5)P_3_ as already reported [23]. Recombinant GST-myotubularin (MTM1) was expressed overnight at 18 °C using 0.8 mM IPTG and purified by affinity chromatography with glutathione Sepharose 4B beads in 20 mM Tris pH 8, 300 mM NaCl, 20 mM β-mercaptoethanol.

Animals: Mice were of C57BL/6 genetic background. Mice were housed in the Anexplo (Toulouse) vivarium according to institutional guidelines. For all experiments, 8- to 14-week-old male or female mice were used. Ethical approval for animal experiments was obtained from the French Ministry of Research in agreement with European Union guidelines (protocol code: APAFIS#3627-2016011516566853).

Human and mouse blood sampling: Human blood samples, collected under citrate from healthy donors, were obtained from the Etablissement Français du Sang (EFS-Toulouse) and immediately processed for experiments. Mouse blood was drawn from the inferior vena cava of anesthetized wild-type (WT) mice, mice deficient for class III PI3-kinase (Vps34^null^) specifically in megakaryocyte and platelets, and mice expressing a heterozygous kinase-dead version of class II PI3-kinase α (PI3KC2α^null^) [24,25]. Platelets were prepared as previously described [26,27].

Preparation of washed platelets from blood: Platelets were isolated from blood through successive centrifugations, as previously described [28]. The washed platelets were then resuspended at 4 × 10^8^/mL in HEPES buffer (140 mM NaCl, 5 mM KCl, 12 mM NaHCO_3_, 5 mM KH_2_PO_4_, 1 mM MgSO_4_, 5 mM glucose, 10 mM HEPES, pH 7.4) for experiments. Mouse platelets were obtained by drawing blood from the inferior vena cava of anesthetized mice into a syringe containing acid citrate dextrose (1:9 *v*/*v*). Washed platelets were prepared as previously described [29], and were resuspended at a concentration of 4 × 10^8^/mL in HEPES buffer containing (140 mM NaCl, 2 mM KCl, 12 mM NaHCO_3_, 0.3 mM NaH_2_PO_4_, 1 mM MgCl_2_, 5.5 mM glucose, 5 mM HEPES, pH 7.4).

Labelling of platelets with fluorescently-tagged PI probes: For in vitro experiments, washed platelets (4 × 10^8^/mL) were fixed with 1.8% *v*/*v* formaldehyde before being labeled with PI probes to detect PIs in the outer leaflet of the plasma membrane (30 min for flow cytometry or 60 min for confocal imaging). The following fluorescently-tagged PI probes, ^mCherry^PH-FAPP1, ^mCherry^PHD-ING2, ^mCherry^PH-PLC-δ1, and ^mCherry^PH-GRP1, were used at a final concentration of 25 µg/mL. PI3P was labeled with ^mcherry^FYVE (15 and 25 µg/mL), ^GFP^AVR L567 (100 and 150 µg/mL), and ^A594^p40PX (25 and 50 µg/mL), or their respective inactive mutants ^mcherry^FYVE_R183A_, ^GFP^AVR L567_ryfr-de-_, and ^A594^p40PX_R57Q_. The anti-PI3P antibody was used at a dilution of 1/100 and the fluorescent secondary antibody at a dilution of 1/300. Phalloidin conjugated to Alexa Fluor 488 or 594 was used at a dilution of 1/1000 concurrently with the probes or the antibody to check membrane integrity. When indicated, cells were permeabilized with 2 mM digitonin in PBS before labeling.

In endocytosis experiments, washed resting platelets were incubated at 37 °C for 30 min with the PI3P probe alone (^mcherry^FYVE 25µg/mL or ^A594^p40PX 50µg/mL), before the addition of ^OregonGreen^ fibrinogen (200 µM) for another 30 min. The platelets were then fixed and analyzed by confocal microscopy.

In in vivo experiments, fluorescent PI3P probes (^mcherry^FYVE, ^mcherry^FYVE_R183A_ (25 µg/mL), or ^GFP^AvrL567 (100 µg/mL)), were intravenously administrated to mice. The blood volume was determined based on the weight of the animals. Whole blood was collected 40 min later and platelets were isolated and fixed for fluorescence analysis or resuspended in HEPES buffer for platelet activation (1 × 10^9^/mL at 37 °C with CRP (1 µg/mL) in the presence of CaCl_2_ (2 mM) and Integrilin (40 µg/mL, GlaxoSmithKline Group (London, UK)) to prevent aggregation. The secretion of platelet granules was collected after two successive centrifugations (1000× *g*) by collecting the supernatant, which was then diluted in Laemmli buffer (100 mM Tris-HCl (pH 6.8), 15% *v*/*v* glycerol, 25 mM DTT, 3% SDS) for Western blotting analysis.

In ex vivo experiments, PI3P probes were added directly to blood for 40 min at 37 °C. Platelets were then collected from PRP and fixed for fluorescence analysis.

Fluorescence analysis by confocal imaging and flow cytometry: To analyze the fluorescence of the labeled platelets, both confocal imaging and cytometry methods were used. The fluorescence of platelets was analyzed using a BD Bioscience Fortessa cytometer and the Diva 8.0 software from Becton Dickinson (San Jose, CA, USA). For confocal microscopy, fixed platelets were spun down on glass slides coated with polylysine and mounted with Mowiol solution. The images were captured using an x63 immersion lens on an LSM 780 confocal scanning microscope and analyzed using Zen Black 2.3 SP1 and ImageJ v1.54c software. In each experiment, a representative sample of three to four randomly selected fields per condition was observed. To quantify the fluorescence intensity, ROI (regions of interest) were isolated for each platelet and analyzed using the ImageJ v1.54c software.

Hydrolysis of the external PI3P by exogenous MTM1 and phospholipase A1 ABH: The hydrolysis of external PI3P by exogenous MTM1 and phospholipase A1 ABH was evaluated by fixing washed platelets and incubating them with the phosphatase myotubularin 1 (MTM1) (5 µg/mL) or the phospholipase A1 ABH (2 µg/mL) for 30 min before staining with the ^mcherry^FYVE probe. The conversion of PI3P into PI by MTM1 or to LysoPI3P by ABH was reflected as a decrease in ^mcherry^FYVE binding to platelets. The specificity of the enzyme effects on external PI3P was confirmed by using the MTM1 inhibitor sodium orthovanadate (0.1 mM) and the inactive mutant ABH_SA_.

To check the phosphatase activity of recombinant MTM1, fluorescent di-C16-bodipyPI3P was used in a buffer containing 50 mM ammonium acetate and 2 mM DTT, at pH 6.0 for 30 min at 30 °C, according to Taylor and Dixon [26]. Lipids were then extracted according to a modified Bligh and Dyer procedure, separated by thin-layer chromatography, visualized, and quantified using a Typhoon image scanner equipped with a blue laser (488 nm) and ImageQuant TL 5.0 software (Appendix A).

Western blotting: Supernatants of activated platelets and platelet lysates were separated on 10% SDS-PAGE, transferred onto a nitrocellulose membrane, and incubated with a GST primary antibody coupled to HRP secondary antibody. The primary antibodies were incubated at 4 °C in TBS buffer containing 1% Tween20 and 1% BSA, as previously described in [28,29].

Statistics: All data are presented as mean ± standard error of the mean (S.E.M.). The data were analyzed using Wilcoxon non-parametric test with the GraphPad Prism 8 software. Statistical significance was set at *p* < 0.05 and indicated as * *p* < 0.05, ** *p* ≤ 0.01, *** *p* ≤ 0.001.

## 3. Results

### 3.1. PI3P Is the Only PI Detected on the Surface of Resting Blood Platelets

Using specific PI-binding domains as fluorescent probes, we investigated the potential presence of PIs in the outer leaflet of blood platelet plasma membranes through imaging approaches [10]. The FYVE domain of the Hrs protein (Hrs-1xFYVE) was used as a specific PI3P probe [27,30] while the Pleckstrin-Homology (PH) domains of FAPP1, PLCδ1, and GRP1 were used to localize PI4P, PI(4,5)P_2_, and PI(3,4,5)P_3_, respectively. The PHD domain of ING2 was used to localize PI5P. These probes, fused with GST and either mCherry or A594 tags, were used on fixed human-washed platelets that were either permeabilized to label all PIs (Figure 1, upper panel) or non-permeabilized to selectively detect PIs that would be present in the outer leaflet of the plasma membrane, called hereafter as external PIs (Figure 1, lower panel). Phalloidin (^A488^phalloidin) was used as a reporter of plasma membrane permeability.

As expected, all PIs were detected by confocal microscopy imaging in permeabilized human platelets; however, with weak labeling of PI5P and PI(3,4,5)P_3_, which are known to be present at very low levels in resting conditions [22,31] (Figure 1, upper panels). In non-permeabilized platelets, the FYVE probe was the only one to decorate the platelet surface (Figure 1, lower panels). This data suggests that PI3P is the sole member of the PIs family to be present in the outer leaflet of the plasma membrane in resting human platelets.

To confirm this observation, we used the ^GST-mCherry^FYVE_R183A_ mutant that does not bind to PI3P, and an unrelated well-characterized PI3P probe, the ^A594^p40PX, as well as its mutant ^A594^p40PX_R57Q_ that no longer binds PI3P. The FYVE and the p40PX probes decorated the plasma membrane of non-permeabilized fixed human-washed platelets while no signal was observed with their inactive mutants (Figure 2A,B). Furthermore, flow cytometry analysis showed a dose-dependent binding of the two PI3P probes in non-permeabilized fixed platelets compared to their inactive mutants (Figure 2D,E).

We also used the ^GST-GFP^AVRL567 probe, which has been shown to bind PI3P through an RFYR-deer motif, and its mutant ^GST-GFP^AVRL567_rfyr-de_, which is unable to interact with PI3P [16,17]. As shown in Figure 2C, the probe decorated non-permeabilized fixed human-washed platelets while its inactive mutant gave no signal. Flow cytometry analysis confirmed the binding of AVRL567 on the surface of non-permeabilized fixed washed human platelets (Figure 2F). Finally, a commercial anti-PI3P antibody was also used and decorated the outer leaflet of fixed human-washed platelets plasma membrane as assessed by confocal microscopy (Figure 3A) and flow cytometry (Figure 3B). As a control, an isotypic antibody gave no signal (Figure 3A,B).

To further check the selectivity of the interaction of the probes with PI3P, competitive binding experiments were performed. The significant decrease of FYVE binding on human-washed platelets preincubated with the anti-PI3P antibody (Figure 3C), as well as of AVRL567 binding on platelets pre-incubated with the non-fluorescent FYVE probe (Figure 3D), indicated a good selectivity of the probes in our experimental setting.

Data obtained with three different specific PI3P probes and the anti-PI3P antibody point to a new pool of PI3P in the outer leaflet of the plasma membrane of resting human-washed platelets that is accessible to protein exhibiting a PI3P binding domain. Interestingly, this external PI3P pool was also present in mouse platelets as assessed by binding experiments with the FYVE and the AVRL567 probes and their mutants (Appendix A).

Altogether, these data strongly suggest the existence of an external PI3P pool in the outer leaflet of the plasma membrane of resting human and mouse washed platelets.

### 3.2. The External PI3P Pool Can Be Metabolized by Exogenous MTM1 Phosphatase and ABH Phospholipase A1 and Involves Class III PI 3-Kinase and Class II PI 3-Kinase α

The concept of plasma membrane phospholipid asymmetry in mammalian cells has been established in erythrocytes and platelets by the addition of exogenous phospholipases acting selectively on the external layer of the plasma membrane under non-lytic conditions [32,33]. Here, we preincubated human platelets with recombinant myotubularin (MTM1), a 3-phosphatase that specifically and efficiently converts PI3P into PI (Appendix A) [34], and with recombinant ABH phospholipase A1, known to selectively transform PI3P into lysoPI3P, a lipid no longer recognized by PI3P-binding domains [35], before incubation with the FYVE probe. Confocal imaging showed a significant decrease in fluorescence intensity following the addition of either MTM1 or ABH (Figure 4A,B). These data were confirmed by measuring the FYVE probe binding by flow cytometry (Figure 4C). Addition of sodium orthovanadate (OV), a potent inhibitor of MTM1 activity [36], abolished the effect of MTM1. Furthermore, the inactive phospholipase A1 mutant, ABH_SA_, had no significant impact on the binding of the FYVE probe. Overall, these data confirm the existence of a PI3P pool in the outer leaflet of the platelet plasma membrane and demonstrate its accessibility to exogenous PI3P metabolizing enzymes.

To evaluate the percentage of PI3P present in this external pool, we performed flow cytometry experiments to compare the PI3P labelling of permeabilized (total PI3P) versus non-permeabilized (external PI3P) platelets. The data indicated that 19.6 ± 5% of the total PI3P is present in the external pool in human platelets and 6.6 ± 0.8% in mouse platelets (Appendix A). It is, however, noteworthy that these values may be overestimated because the permeabilization treatment could have removed some PI3P. Moreover, we observed that human platelet activation by collagen-related peptide (CRP) significantly increased the binding of the PI3P probe on their surface (Appendix A). The binding of the PI3P probe on the surface of mouse platelets also showed a tendency to increase following activation by CRP (Appendix A). This increase may be attributed to a contribution of the secretory granules or to the open canalicular system, which becomes accessible following platelet stimulation.

Class II PI3-kinase α and class III PI3-kinase (Vps34) are known to contribute to PI3P production in platelets [21,24,25]. Therefore, we checked their potential contribution to the formation of the surface PI3P pool by taking advantage of platelets deficient (Vps34^null^) or inactivated (PI3KC2α^KI^) for these kinases. Using a mouse model deficient for class III PI3K (Vps34) specifically in the megakaryocyte/platelet lineage (Vps34^null^) [25], we found by confocal imaging analysis that the staining of the FYVE probe on fixed non-permeabilized washed platelets was significantly reduced in the absence of class III PI3K (Figure 5). Moreover, platelets from mice expressing a heterozygous kinase-dead class II PI 3-kinase α [24] also showed a significantly decreased staining of the external pool of PI3P by the FYVE probe (Figure 5). These data indicate a contribution of these two isoforms of PI 3-kinases in the production and/or location of the platelet external PI3P pool.

### 3.3. Interaction of PI3P Binding Domains with Platelets in Whole Blood In Vivo in Mouse and Ex Vivo in Human

We then investigated whether PI3P-binding domains would interact with the external PI3P pool on platelets in whole blood, both in vivo in mouse or ex vivo in freshly drawn human blood. As shown in Figure 6A, peripheral staining was observed on platelets from mice injected with the FYVE probe, with some dots inside platelets. Conversely, the fluorescence intensity of the inactive FYVE^R183A^ mutant on platelets was very weak. Similarly, following injection of the AVRL567 probe in the mouse blood circulation, platelets were positive with, however, more punctiform staining, suggesting an uptake of the probe by platelets. The AVRL567^rfyr-de-^ inactive mutant could not be used for in vivo experiments as we failed to concentrate it sufficiently for injection in the bloodstream. To circumvent this drawback, we compared the in vivo platelet AVRL567 labelling in WT and in Vps34^null^ mice, which have a reduced amount of PI3P in the outer leaflet of the plasma membrane. Consistent with the in vitro observation (Figure 5), a significant decrease in AVRL567 staining was observed in Vps34^null^ deficient platelets (Figure 6A).

Considering the punctate labeling of the AVRL567 probe observed by confocal imaging, we then explored the potential endocytosis of PI3P-binding domains in platelet granules in vivo. The AVRL567 probe was injected into the mouse blood circulation, and after 40 min, platelets were isolated, resuspended in HEPES buffer, activated by CRP, and the proteins secreted from platelet granules were collected and analyzed by Western blotting using an anti-GST antibody. Interestingly, the AVRL567 probe was detected in the secretome of CRP-activated wild-type platelets, while it was weakly detected in CRP-activated Vps34^null^ platelets secretome (Figure 6B). These results strongly suggest that, in vivo, AVRL567 binds to the platelet external PI3P pool, is endocytosed, and transported to platelet granules. To confirm these data, similar experiments were performed using human blood, in which two different PI3P binding domains, FYVE and AVRL567, and their inactive mutants were added ex vivo. After 40 min at 37 °C, the FYVE and AVRL567 probes were found in 20% and 30% of human platelets, respectively (Figure 6C). Conversely, their inactive mutants, FYVE_R183A_ and AVRL567_rfyr-de-_, were hardly detectable (Figure 6C). In parallel, a fraction of platelets was stimulated by CRP and the platelet secretome was analyzed by Western blotting. The AVRL567 domain was strongly detected in the CRP-activated platelet secretome, while its inactive mutant was barely detectable (Figure 6D). Similarly, but to a lower extent, the FYVE domain, but not its inactive mutant, was found in the platelet secretome (Figure 6D).

### 3.4. Endocytosis of the PI3P-Binding Domains to Platelet α-Granules

Fibrinogen is a plasma protein known to be internalized by circulating platelets and stored in α-granules through an α_IIb_β_3_-mediated clathrin-dependent endocytosis process [37,38]. Fluorescent fibrinogen is an established marker of platelet α-granules [37,38,39]. To investigate the potential uptake of FYVE and p40PX domains in α-granules, the probes or their inactive mutants were added to resting washed human platelets at 37 °C and 30 min later, ^OregonGreen^fibrinogen was added to the platelet suspension for another 30 min. Confocal microscopy shows that the FYVE and the p40PX probes were internalized, as evidenced by their punctiform intracellular fluorescence, and colocalized with fluorescent fibrinogen in α-granules (Figure 7A,B). This colocalization was highlighted by linescan analysis and measured by the Pearson’s coefficient (R = 0.656 and R = 0.779 for the FYVE and the p40PX probes, respectively) (Figure 7A,B, right panels). The FYVE_R183A_ and p40PX_R57Q_ mutants did not interact with platelets and were not internalized while at the same time, fibrinogen was taken up into α-granules (Figure 7A,B, lower panels).

These results show that the interaction of PI3P-binding domains to the platelet external PI3P pool triggers internalization of the domains that eventually accumulate in α-granules and can be secreted following platelet activation.

## 4. Discussion

PIs are widely known as intracellular lipid messengers that recruit effector proteins to the cytoplasmic side of cell membranes to regulate key biological processes such as signal transduction, cytoskeleton reorganization, and intracellular trafficking [1,2]. However, the question of whether PIs are present in the outer leaflet of the cell plasma membrane is a relatively recent one and is still an open topic. In 2010, Kale et al. [16] demonstrated the presence of PI3P in the outer leaflet of plant cells plasma membrane, where it plays a role in the intracellular translocation of fungal pathogens effectors that have a PI3P-binding domain. The idea that external pools of PIs may play a role in the communication between cells and their extracellular environment has been further explored recently [18,19]. In this study, by using binding domains specific to each PI coupled to a fluorescent and/or a GST tag, we have shown that PI3P is the only PI detectable in the outer leaflet of the plasma membrane of human and mouse resting blood platelets. The use of three unrelated specific PI3P probes, including FYVE, PX, and AVRL567 domains and their inactive mutants, as well as an anti-PI3P antibody, conclusively demonstrated the presence of a specific external PI3P pool in platelets. Under these conditions, PI(4,5)P_2_ or PI4P, which are much more abundant than PI3P in resting platelets, were not found to be present, arguing for a specific role of PI3P on the outer leaflet of the platelet plasma membrane. Moreover, the use of two specific PI3P converting enzymes (the 3-phosphatase MTM1 [34,36] and the phospholipase ABH [35]) impaired the recognition signal of the external PI3P pool by the specific biosensors, indicating that this PI3P pool was indeed accessible to and hydrolysable by exogenous metabolizing enzymes. These data indicate that proteins with a domain of recognition of PI3P can directly interact with this external pool of PI3P on platelets.

Importantly, we found that class II PI 3-kinase α and class III PI 3-kinase (Vps34) play a role in the production of the external PI3P pool in platelets. However, the mechanism by which PI3P reaches the outer leaflet of the platelet plasma membrane remains an open and challenging question. It is possible that this lipid is produced in the plasma membrane or transported to the membrane via recycling endosomes or possibly exosomes, and then externalized by a putative flippase. Another possibility is that this external PI3P pool is formed during platelet generation by megakaryocytes. Indeed, it has been shown that during megakaryocyte maturation, PI3P derived from Vps34 is involved in the formation of demarcation membranes and proplatelet production through late endosomes/lysosomes [40]. The fusion of PI3P-containing late endosomes/lysosomes and the dynamic exchanges and traffic of the endo-lysosomal membrane system with the developing demarcation membranes in megakaryocytes may be the origin of the platelet external PI3P pool identified in our study. Further research is needed to fully understand how the platelet external PI3P is formed and maintained.

Another important question is the function of this external PI3P pool in the communication of platelets with the extracellular environment. We have discovered that once bound to the platelet external PI3P pool, proteins containing PI3P-binding domains, but not their inactive mutants, are internalized in platelets both in vivo in mouse and ex vivo in human blood and can be secreted following platelet activation by CRP. Unlike the fungal plant pathogen effectors, which enter host cells through PI3P-binding and escape the endolysosomal pathway, PI3P-binding domain containing proteins in platelets are largely directed to α-granules. Once stored in α-granules, PI3P-binding domains can be secreted following platelet activation. This is the first example of protein uptake and storage in platelet α-granules through binding to a PI on the external leaflet of the plasma membrane. Whether internalization of plasma proteins containing PI3P-binding domains by platelets and sorting to α-granules play a physiological role or is part of a defense mechanism to eliminate toxic proteins, such as mammalian infective fungi effectors [41], from the plasma is currently unknown. This finding also opens the possibility that PI3P-binding domains may be used to target potential therapeutic proteins in platelet α-granules before transfusion.

In conclusion, this study provides a new understanding of the various roles of PIs in cell biology. The discovery of an external PI3P pool in the plasma membrane of blood platelets, which directs binding cargoes to α-granules, opens up new possibilities for research in platelet biology and function.

## Figures and Tables

**Figure 1 biomolecules-13-00583-f001:**
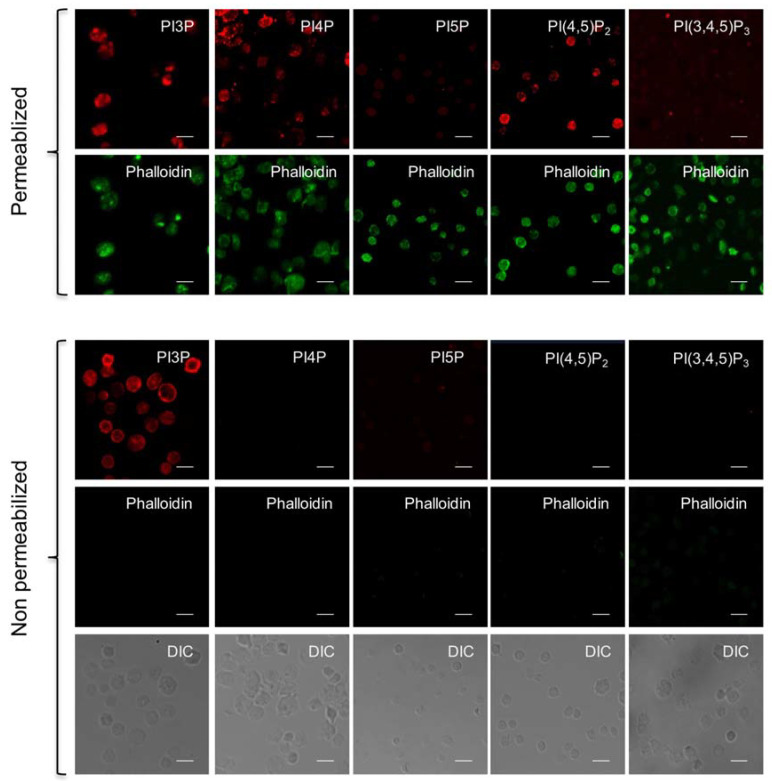
PI3P is the sole phosphoinositide detectable on the surface of fixed, non-permeabilized human blood platelets. Fixed, washed resting human platelets were either permeabilized (for total detection, upper panels) or non-permeabilized (for surface detection only, lower panels) and PIs were labeled by incubation for 60 min with appropriate fluorescent-specific probes at a concentration of 25 µg/mL. The FYVE domain of Hrs was used to label PI3P, the PH domain of FAPP1 for PI4P, the PHD domain of ING2 for PI5P, the PH domain of PLC-δ1 for PI(4,5)P_2_, and the PH domain of GRP1 for PI(3,4,5)P_3._ The F-actin probe, ^488^Alexa -conjugated phalloidin, was used to monitor and control cell permeabilization, and differential interference contrast (DIC) was used to detect the presence of non-permeabilized platelets in the field (lower panel). Confocal microscopy images shown are representative of 3 to 5 independent experiments with similar results. Scale bar: 5 µm.

**Figure 2 biomolecules-13-00583-f002:**
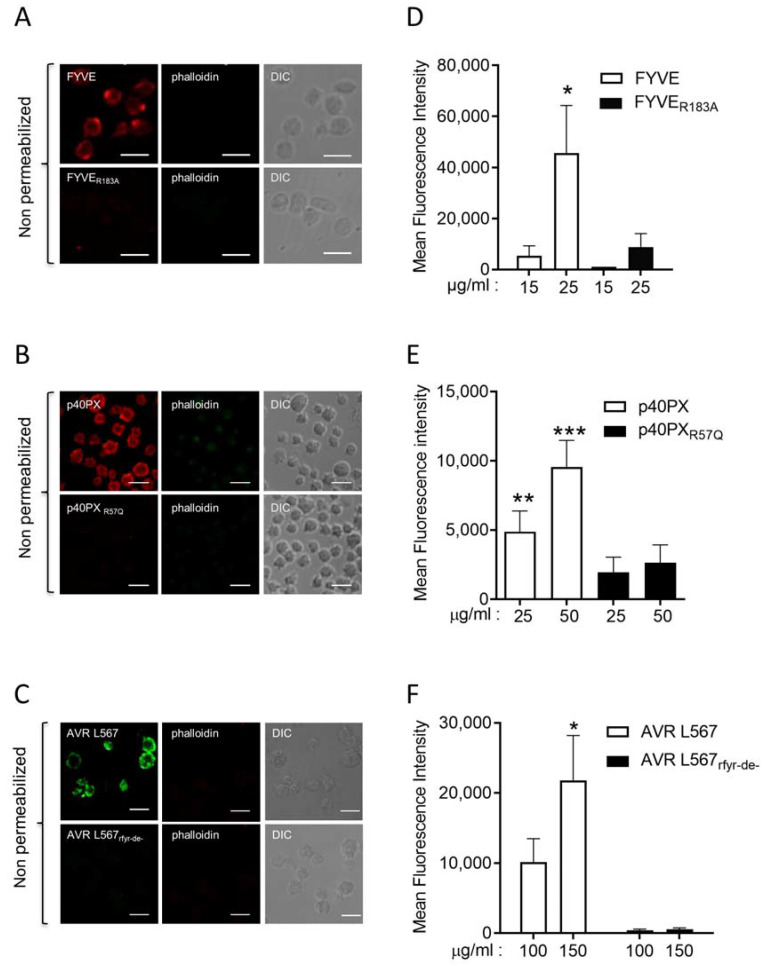
Three unrelated PI3P-binding domains recognize an external pool of PI3P on resting human platelets. Fixed, non-permeabilized, resting human platelets were incubated with three distinct fluorescent specific PI3P probes and their inactive mutants, and analyzed using confocal microscopy, as in Figure 1. Images shown are representative of 5 independent experiments with FYVE or FYVE_R183A_ (25 µg/mL) (**A**) and p40PX or p40PX_R57Q_ (50 µg/mL) (**B**), and 3 independent experiments with AVR L567 or AVR L567_rfyr-de-_ (100 µg/mL) (**C**). The absence of ^488^Alexa-conjugated phalloidin fluorescence confirms the integrity of the plasma membrane. PI3P labeling was also quantified in fixed, non-permeabilized, resting human platelets using the three different probes and their inactive mutants by flow cytometry (**D**–**F**). Results are presented as mean ± SEM of fluorescence intensity from 5 independent experiments with the FYVE probes and 3 independent experiments with p40PX and AVR L567 probes. Statistical analysis using Wilcoxon test compared each PI3P probe to its mutant used at the same concentration (* *p* ≤ 0.05, ** *p* ≤ 0.01, *** *p* ≤ 0.001).

**Figure 3 biomolecules-13-00583-f003:**
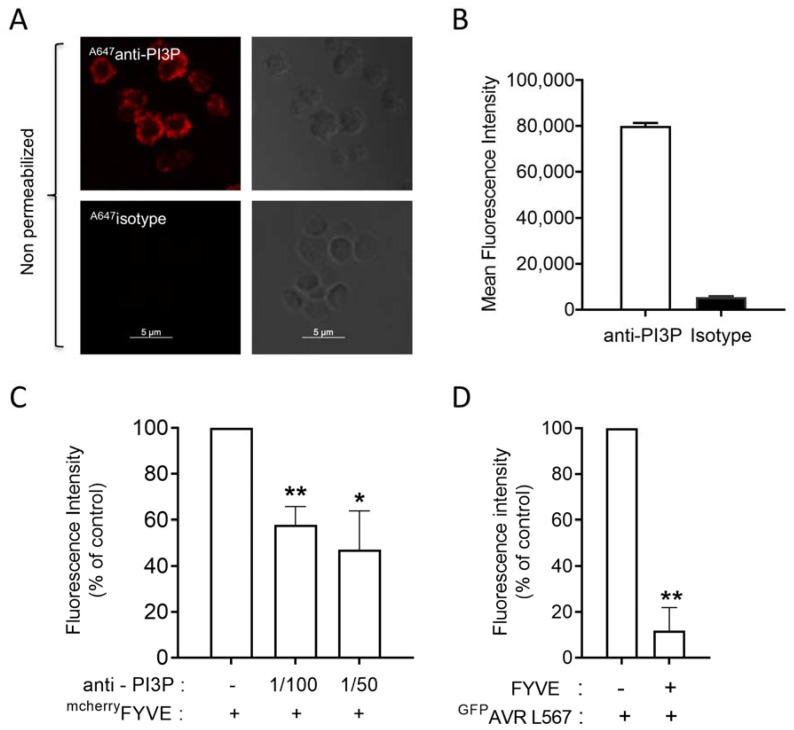
The anti-PI3P antibody recognizes the external PI3P pool on non-permeabilized human platelets and competes with the FYVE probe labeling. Confocal microscopy was used to visualize the external PI3P pool on fixed human-washed platelets, labeled with the anti-PI3P commercial antibody coupled to a secondary ^594^Alexa anti-mouse antibody (**A**, upper panel). An isotopic antibody was used as a control ((**A**), lower panel). Scale bar: 5µm. The images shown are representative of 3 independent experiments. Flow cytometry was also used to detect PI3P with the anti-PI3P commercial antibody (**B**). The graph represents the mean ± SEM of 3 independent experiments. The isotopic antibody was used as a control. The anti-PI3P antibody (used at 1/100 and 1/50) was able to compete with the binding of the fluorescent FYVE probe to the platelet external PI3P pool, as assessed by flow cytometry (**C**). Furthermore, pre-incubation of the non-fluorescent FYVE probe decreased AVR L567 probe binding to the human-washed platelet external PI3P pool (**D**). The results are expressed as percentage of control (FYVE for C and AVR L567 for D) and are mean ± SEM of 3 independent experiments (Wilcoxon test compared to control, * *p* ≤ 0.05, ** *p* ≤ 0.01).

**Figure 4 biomolecules-13-00583-f004:**
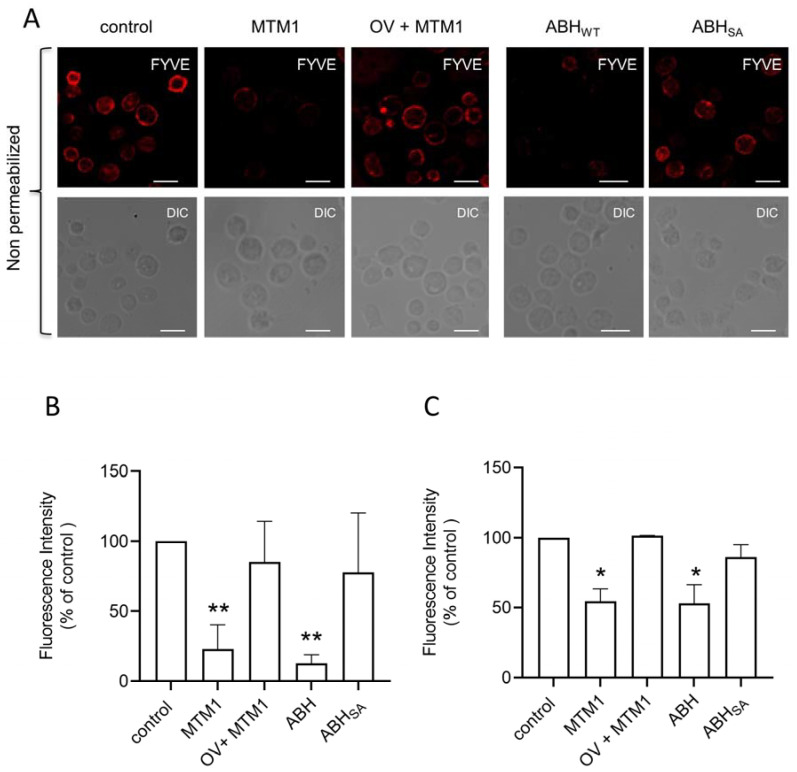
The external PI3P pool on platelets can be metabolized by exogenous 3-phosphatase MTM1 and phospholipase ABH. The external PI3P pool was labeled with the FYVE probe (25 µg/mL) in fixed human-washed platelets pretreated or not (control) with the recombinant 3-phosphatase MTM1 (MTM1, 5 µg/mL) in the presence or absence of sodium orthovanadate (OV) used as an MTM1 phosphatase inhibitor, or with the recombinant phospholipase A1 (ABH, 8 µM) or its inactive mutant (ABH_SA_, 8 µM) (**A**). Confocal images shown are representative of 3 independent experiments. Scale bar: 5 μm. The fluorescence intensity in 200 platelets from the 3 independent experiments was quantified using the ImageJ software (**B**). Flow cytometry analysis of the external PI3P pool labeled with the FYVE probe on fixed non-permeabilized human-washed platelets that were treated or not treated (control) with MTM1 or ABH as in (**A**) was also performed (**C**). Results are expressed in percentage of control (FYVE probe fluorescence intensity in control platelets) and are the mean ± SEM of 3 independent experiments. Wilcoxon test compared the different conditions to control, * *p* ≤ 0.05, ** *p* ≤ 0.01.

**Figure 5 biomolecules-13-00583-f005:**
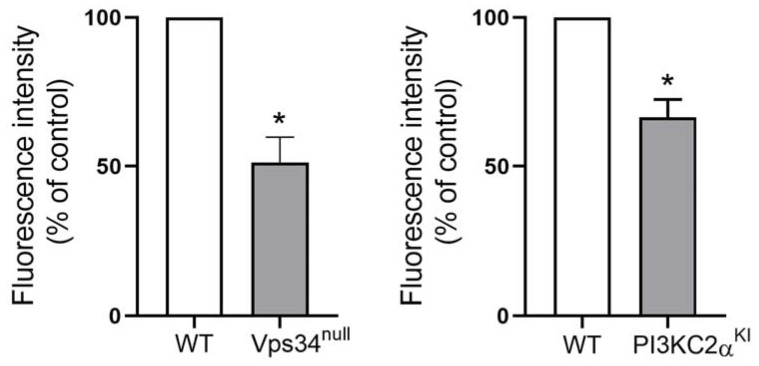
Class III PI 3-kinase (Vps34) and Class II PI 3-kinase α contribute to the external PI3P pool on platelets. To investigate the role of class III PI3-kinase (Vps34) and class II PI3-kinase α in the formation of the external PI3P pool on platelets, platelets were isolated from wild type (WT), Vps34-deficient megakaryocyte/platelet (Vps34^null^), and heterozygous kinase dead class II PI 3-kinase α (PI3KC2α^KI^) mice. These fixed washed platelets were then incubated with the FYVE probe (25 µg/mL) and analyzed using confocal imaging. The fluorescence intensity was quantified using the ImageJ software from a total of 280 WT, 260 Vps34^null^, and 215 WT and 89 PI3KC2α^KI^ in 3 independent experiments. The results are expressed as a percentage of control (WT platelets). Wilcoxon test compared the different conditions to control, * *p* ≤ 0.05.

**Figure 6 biomolecules-13-00583-f006:**
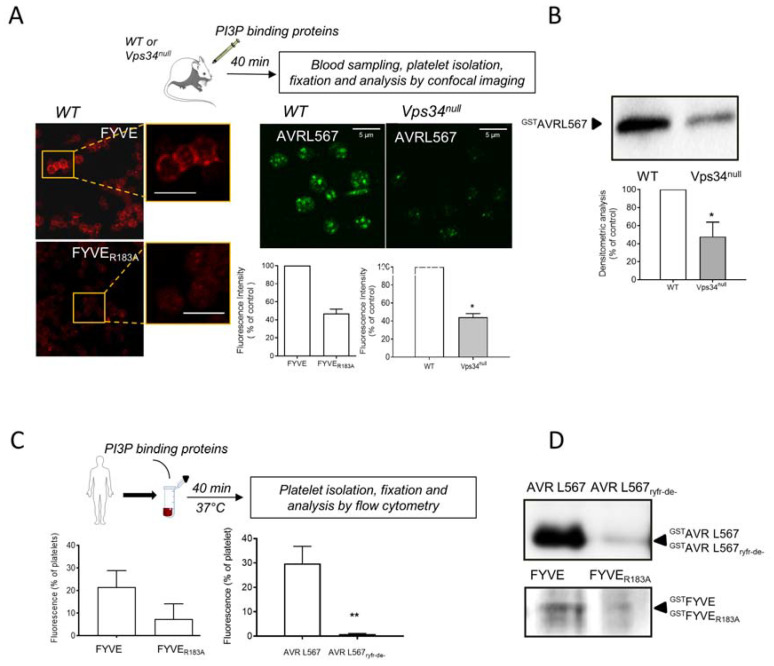
Interaction of PI3P-binding domains with platelets in vivo and ex vivo. The FYVE probe and its mutant FYVE^R183A^ (25 µg/mL, in red) were injected into the blood circulation in WT mice (**A**). The AVR L567 probe (100 µg/mL, in green) was also injected into the blood circulation of WT and Vps34-deficient megakaryocyte/platelet (Vps34^null^) mice. Blood was collected 40 min post-injection and platelets were isolated and fixed before fluorescence analysis by confocal imaging (**A**). The images shown are representative of 3 independent experiments. Scale bar: 5 μm. The fluorescence intensity per platelet (n = 300) was quantified using the ImageJ software ((**A**), lower panels) and the results are expressed as a percentage of control (FYVE probe fluorescence in WT mice or the AVR L567 fluorescence in WT mice for Vps34^null^ mice). In addition, platelets collected after injection of the AVR L567 probe were also activated in vitro in HEPES buffer with Collagen Related Peptide (CRP, 1 µg/mL), and the secretome of activated platelets was collected by centrifugation to analyze the presence of the AVR L567 probes by Western blotting with an anti-GST antibody (**B**). A representative image of the Western blotting from 3 independent experiments is shown. The relative band intensity was quantified using the Image Lab 6.1.0.7-2 software (Bio-Rad) and results are expressed as a percentage of control (AVR L567 band intensity in WT mice platelet secretome). The FYVE and AVR L567 probes and their inactive mutants were added to freshly collected human blood and after 40 min at 37 °C, platelet-rich plasma was obtained by centrifugation. After dilution in a 10-fold volume of HEPES buffer, platelets were collected by centrifugation and fluorescence was analyzed by flow cytometry (**C**). The results are presented as the percentage of fluorescent platelets, with the mean ± SEM calculated from 3 independent experiments. Additionally, the collected platelets were activated in vitro in HEPES buffer with Collagen Related Peptide (CRP, 1 µg/mL) and their secretome was then harvested by centrifugation and analyzed using Western blotting with an anti-GST antibody (**D**). A representative image from 3 independent experiments is presented. Wilcoxon test compared the different conditions to control (**A**,**B**) or FYVE and AVR L567 (**C**), * *p* ≤ 0.05, ** *p* ≤ 0.01.

**Figure 7 biomolecules-13-00583-f007:**
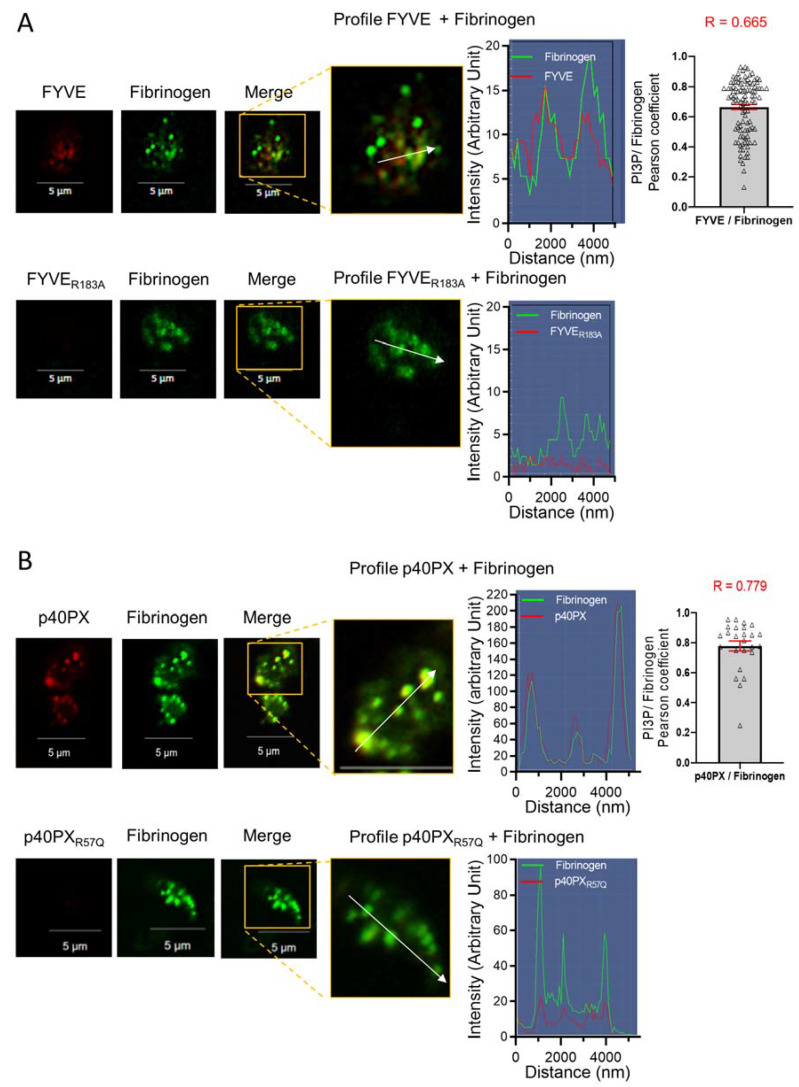
Exogenous PI3P-binding domains are endocytosed by platelets and accumulate in α-granules. Washed resting human platelets were incubated at 37 °C for 30 min with the FYVE probe or its mutant FYVE_R183A_ (**A**), as well as the p40PX probe and its mutant p40PX_R57Q_ (**B**), before addition of ^OregonGreen^fibrinogen (green), a well-established α-granules marker, for another 30 min. The colocalization of PI3P and fibrinogen is shown in yellow on the merge channel and was measured using the Pearson coefficient (R) in ImageJ software (from 106 and 26 platelets for FYVE and PX probes, respectively). The linescans (right panels) show the distribution of fluorescence along the white arrow. These images are representative of 3 independent experiments.

## Data Availability

The data presented in this study are available in Appendix A here.

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
