# Peer review of "Exploring the Role of PI3P in Platelets: Insights from a Novel External PI3P Pool"

_biomolecules, 2023, doi:10.3390/biom13040583_

Round 1

Reviewer 1 Report

Suggestions for authors:

1. in the Results section and all Figure legends/Supplementary Figure legends: please specify each time if platelets were fixed (and/or washed) (eg lanes 249, 252, 254, 271, 276,...). This is important because fixation could somehow alter the membrane and PI distribution within the membrane. Hopefully, in the in vivo experiments in mouse, the injected PH probe binds to unmanipulated (ie non fixed) platelets, suggesting that fixation does not seem to alter PI distribution in platelets.

2. minor corrections:

-l 439: platelets (e is missing)

-l 365-369: this paragraph belongs to the Figure 6 legend (and not to the Results section) and thus should be written with the font size of the legend (not of the results)

-Figure 5 legend: inverse "class III PI 3-kinase (Vsp34)" and "class II PI 3-kinase alpha" in the title and the first sentence, because the graph shows first (on the left) class III PI 3-kinase (Vsp34), and then (on the right) the class II PI 3-kinase alpha

-Figure 7B: red (mainly, but also green) linescans are too weak, more intense colors should be presented      

Reviewer 2 Report

This study demonstrates the existence of a pool of PI3P on the outer leaflet of the plasma membrane in resting platelets. The combined use of a wide array of external recombinant  probes and enzymes in their active form or inactive mutants makes the observations strong and solid. Studies on transgenic mice document the involvement of both class II and class III PI3Ks in external PI3P generation. The authors also conclude that the external PI3P pool triggers the internalization of interacting proteins and their targeting into platelet alpha-granules. Overall the study is timely and interesting, but some results could be implemented.

 Major comments

 1. The external pool of PI3P has been documented exclusively in resting platelets. It would be interesting to evaluate possible changes consequent to platelet activation. These changes could be easily quantified through a flow cytometry approach. This analysis is important to understand the dynamics of this novel eternal pool of PI3P, but it will also help to improve the study as it may provide important clues  in order to solve some still open questions such as the mechanism by which PI3P reaches the outer leaflet of the plasma membrane and  the possibility, suggested by the analysis of the secretome, that PI3P (and bound internalized ligand) can be stored into alpha-granules and released during secretion (see also below).

 2. An important  missing piece of information is whether the external PI3P represents a considerable or a negligible  percentage of the total platelet PI3P. Quantification of PIP3 (absolute amount or even just fluorescent binding of specific probes in permeabilized cells) before and upon removal of the external pool with  myotubulin or ABH phospholipase may provide interesting information.

 3. The conclusion that PI3P binding proteins are internalized mainly derives from their recovery in the supernatant of stimulated platelets. The authors should try do provide evidence that exclude that platelet stimulation leads to the dissociation of membrane bound fluorescent probes, thus producing their recovery in the supernatant, maybe by altering the dynamics of PI3P.

 4. To tackle the problem of the origin of the external pol of PI3P, have the authors  tried to evaluate recovery of external PI3P expression (as fluorescent probes binding ) over time upon removal of the preexisting PIP3 molecules by phosphatase or phospholipase treatments? Again, the analysis on stimulated platelets under these conditions may be informative.

 5. The first sentence at page 13 (lines 365-369) is unclear and it seems to anticipate results discussed afterwards.

Author Response

We would like to thank the reviewer for his/her evaluation of our manuscript and we appreciated the interesting points raised to further improve the manuscript.

  1. The external pool of PI3P has been documented exclusively in resting platelets. It would be interesting to evaluate possible changes consequent to platelet activation. These changes could be easily quantified through a flow cytometry approach. This analysis is important to understand the dynamics of this novel eternal pool of PI3P, but it will also help to improve the study as it may provide important clues  in order to solve some still open questions such as the mechanism by which PI3P reaches the outer leaflet of the plasma membrane and  the possibility, suggested by the analysis of the secretome, that PI3P (and bound internalized ligand) can be stored into alpha-granules and released during secretion (see also below).

We agree with the reviewer on this point, and we have now included data showing that there is a significant increase in the binding of the PI3P probe on the surface of platelets following activation by collagen-related peptide (new Supplemental Figure 3B). This observed increase may be attributed to a contribution of the secretory granules or to the open canalicular system, which becomes accessible following platelet stimulation. This result is now commented in text (p11).

  1. An important missing piece of information is whether the external PI3P represents a considerable or a negligible percentage of the total platelet PI3P. Quantification of PIP3 (absolute amount or even just fluorescent binding of specific probes in permeabilized cells) before and upon removal of the external pool with  myotubulin or ABH phospholipase may provide interesting information.

We agree with the reviewer that this would be an important piece of information to provide. The experiment suggested by the reviewer is a very good idea. However, we believe that myotubularin or ABH phospholipase do not hydrolyze 100% of the external PI3P pool (see Fig.4). To overcome this limitation, we have performed flow cytometry experiments that compared the PI3P labelling of permeabilized (total PI3P) versus non-permeabilized (external PI3P) platelets. The data we obtained indicate that about 19.6% of the total PI3P is present in the external pool in human platelets and about 6.6% in mouse platelets. However, this estimation may be overestimated because the permeabilization treatment could have removed some PI3P. We have included these data in a new Supplemental Figure 3A and B and we interpret them with caution in text (p10-11).

  1. The conclusion that PI3P binding proteins are internalized mainly derives from their recovery in the supernatant of stimulated platelets. The authors should try do provide evidence that exclude that platelet stimulation leads to the dissociation of membrane bound fluorescent probes, thus producing their recovery in the supernatant, maybe by altering the dynamics of PI3P.

The conclusion that PI3P binding proteins are internalized is based on their recovery in the supernatant of stimulated platelets, as well as from colocalization experiments of two different PI3P probes with fibrinogen, which is well known to be internalized by platelets and stored in a-granules (Fig.7). Additionally, we have performed some experiments that demonstrate the internalization of the AVRL567 probe in platelets and its good colocalization with P-selectin, a well-established marker of a-granules (see figure to the reviewer below). Moreover, as discussed in response to point 1 from the reviewer, we can now exclude the possibility that platelet stimulation leads to the dissociation of membrane-bound fluorescent probes, because there is an increased binding instead. Overall, our results from different approaches support the conclusion that PI3P binding proteins are internalized in a-granules by platelets.

Figure 1 to reviewer: Washed resting human platelets were incubated at 37°C with the AVRL567 or its mutant AVRL567rfy-de- probes (green) for 30 min. The platelet plasma membrane was then labelled with an anti-CD42b antibody (red) (left panels). These representative confocal images show that the AVRL567, but not its mutant, was internalized in platelets. In the right panel, washed resting human platelets were incubated at 37°C with the AVRL567 probe (green) for 30 min, fixed, permeabilized and labeled with an anti-P-selectin antibody (red). The colocalization of the AVRL567 probe and P-selectine appears in yellow on the merge channel. Results shown are representative of two independent experiments.

  1. To tackle the problem of the origin of the external pol of PI3P, have the authors  tried to evaluate recovery of external PI3P expression (as fluorescent probes binding ) over time upon removal of the preexisting PIP3 molecules by phosphatase or phospholipase treatments? Again, the analysis on stimulated platelets under these conditions may be informative.

This is a very interesting and challenging point. We attempt to evaluate the recovery of external PI3P, but we were unable to interpret the experiment’s data because the treatment with myotubularin or ABH phospholipase requires some time (at least 30 min) and after removing the enzyme by centrifugation, we were unable to maintain the platelets in good conditions for a sufficient amount of time to observe a recovery, or it could be that this pool of PI3P originates from megakaryocytes demarcation membranes. Therefore, we prefer to remain prudent on this point in this manuscript.

  1. The first sentence at page 13 (lines 365-369) is unclear and it seems to anticipate results discussed afterwards.

As also pointed by reviewer 1 this sentence in fact belongs to the legend of Figure 6 and should be written with the font size of the legend.

Round 2

Reviewer 2 Report

The authors have adequately answered to my previous concerns. I have no further requests.